# Photodynamic Suppression of Enterococcus Faecalis in Infected Root Canals with Indocyanine Green, Trolox^TM^ and Near-Infrared Light

**DOI:** 10.3390/pharmaceutics15112572

**Published:** 2023-11-02

**Authors:** Markus Heyder, Markus Reise, Julius Burchardt, André Guellmar, Julius Beck, Ulrike Schulze-Späte, Bernd Sigusch, Stefan Kranz

**Affiliations:** Department of Conservative Dentistry and Periodontology, University Hospital Jena, An der Alten Post 4, 07743 Jena, Germany; markus.heyder@med.uni-jena.de (M.H.); markus.reise@med.uni-jena.de (M.R.); julius.burchardt@acor.de (J.B.); andre.guellmar@med.uni-jena.de (A.G.); julius.beck@med.uni-jena.de (J.B.); ulrike.schulze-spaete@med.uni-jena.de (U.S.-S.); bernd.w.sigusch@med.uni-jena.de (B.S.)

**Keywords:** aPDT, biofilm, disinfection, flare-up, microflora, photosensitizer, photodynamic therapy, root canal infection, root canal treatment, secondary infection

## Abstract

Recently, our group showed that additional supplementation of Trolox™ (vitamin E analogue) can significantly enhance the antimicrobial photodynamic effect of the photosensitizer Indocyanine green (ICG). Up to now, the combined effect has not yet been investigated on Enterococcus faecalis in dental root canals. In the present in vitro study, eighty human root canals were inoculated with *E. faecalis* and subsequently subjected to antimicrobial Photodynamic Therapy (aPDT) using ICG (250, 500, 1000 µg/mL) and near-infrared laser light (NIR, 808 nm, 100 Jcm^−2^). Trolox™ at concentrations of 6 mM was additionally applied. As a positive control, irrigation with 3% NaOCl was used. After aPDT, root canals were manually enlarged and the collected dentin debris was subjected to microbial culture analysis. Bacterial invasion into the dentinal tubules was verified for a distance of 300 µm. aPDT caused significant suppression of *E. faecalis* up to a maximum of 2.9 log counts (ICG 250 µg/mL). Additional application of Trolox^TM^ resulted in increased antibacterial activity for aPDT with ICG 500 µg/mL. The efficiency of aPDT was comparable to NaOCl-irrigation inside the dentinal tubules. In conclusion, ICG significantly suppressed *E. faecalis*. Additional application of Trolox^TM^ showed only minor enhancement. Future studies should also address the effects of Trolox^TM^ on other photodynamic systems.

## 1. Introduction

Periradicular infections of the tooth are mainly caused by pathogenic microorga-nisms within the root canal system. Often the inflammatory process is also associated with periforaminal and foraminal resorptions [1]. While in primary cases a variety of different Gram-positive and Gram-negative bacteria are found, secondary disease is characterized by a rather limited and selective number of bacterial species [2]. In cases with reported post-therapy pain and infection, Enterococcus faecalis (*E. faecalis*) is one of the most frequently detected endodontic species [3,4].

Responsible for the high prevalence of *E. faecalis* are mechanisms that allow an adaption of the species to the specific intracanal environment [5]. Unlike other pathogens, *E. faecal* is can also resist many disinfectants commonly applied during chemo-mechanical root canal treatment. In this context, it was shown that instrumentation and irrigation with sodium hypochlorite (NaOCl) is often insufficient in removing *E. faecalis* completely [6,7,8]. As also observed, *E. faecalis* can withstand temporary intracanal dressings with anti-infective agents such as calcium hydroxide for an extended period of time [7,9]. Furthermore, the species is resistant to many antibiotics, including erythromycin, azithromycin, tetracycline and clindamycin [3,10,11]. Due to the ability to penetrate deep into the dentinal tubules, *E. faecalis* is often well protected from most of the antimicrobial measures commonly applied during root canal treatment [3,7,12,13].

To control recurrent and president endodontic diseases, further efficient antiseptic measures are still needed. In this context, antimicrobial Photodynamic Therapy (aPDT) has promising characteristics [14,15,16,17,18,19]. In aPDT, exposure of a so-called photosensitizer (PS) to light of a specific wavelength results in the formation of reactive oxygen radicals which cause lethal damage to surrounding bacterial cells [20]. It is suggested that laser-based strategies are beneficial and promising tools, showing efficacy in reducing the microbial load. But, the outcome in comparison to conventional endodontic disinfection methods is still controversially discussed [21,22,23,24].

In aPDT, an adequate availability of PS and light is crucial and has a strong impact on the antibacterial efficiency. Because root canal anatomy is a limiting factor, new strategies for enhancing aPDT have been introduced. In this context, it was proven that the use of special carrier systems for a more sufficient delivery of PS molecules is one promising measure [25]. As also shown by our group, PS delivered by liposomes and invasomes are highly efficient in suppressing *E. faecalis*, but still need adaption to the endodontic system [15,16]. Besides the use of special carriers, the application of light at higher wavelengths is another strategy to increase the efficiency of aPDT.

Due to its physical nature, long-wavelength light can penetrate deeper into tissue. In this regard, near-infrared light (NIR), which is in the range of 700 to 1350 nm shows high practicability [26]. As reported by various authors, NIR light (808 nm) in combination with the photosensitizer Indocyanine green (ICG) also presents favorable antibacterial characteristics [27,28,29]. Unlike aPDT, illumination of ICG with NIR light causes the surrounding temperature to increase strongly which results in a thermal antimicrobial effect [30,31]. Therefore, ICG is also often applied in antimicrobial Photothermal Therapy (aPTT) [32,33]. Besides the thermal properties, ICG also presents photodynamic activity [34,35].

However, extensive heating of the root canal system can cause thermal damage to the periodontium and must be avoided strongly. As shown by our group, additional supplementation of the antioxidant Trolox™ (vitamin E analogue) can significantly enhance the antimicrobial photodynamic effect of ICG without extensive heating of the surrounding [34]. It was suggested that the enhancing effect arises from an excessive formation of antioxidant radicals due to either spontaneous autooxidation or radical scavenging during the photodynamic reaction [36]. It was also realized that antioxidants can also serve as a substrate for photosensitizers in the excited triplet state. This causes the formation of radical photosensitizer anions and antioxidant radicals which might further react with residual oxygen leading to the formation of superoxide and antimicrobial active radical anions [37].

Up to now, the antibacterial effect of a combined photodynamic treatment of ICG, Trolox™ and NIR light on *E. faecalis* inside dentinal tubules has not yet been analyzed. Therefore, the present in vitro study aimed to investigate the antimicrobial effect of ICG in combination with Trolox™ and NIR laser light on *E. faecalis* in artificially infected human dental root canals. The null hypothesis claims that there is no enhancing effect of aPDT with ICG after additional application of Trolox™.

## 2. Materials and Methods

### 2.1. Cultivation of Enterococcus Faecalis

*E. faecalis* (DSMZ 20376) was cultivated in Schaedler fluid media (CM0497, Oxoid Ltd., Hampshire, UK) for 24 h. Afterwards, bacteria were pelleted (4000  rpm, 8  min) and washed twice with PBS. Cells were resuspended in PBS to an optical density (OD 546  nm) of 0.5 (106–107 bacterial cells/mL). The received bacterial solution was immediately used for root canal inoculation.

### 2.2. Preparation of Root Specimens for E. faecalis Inoculation

For the present study, 80 freshly extracted third molars were used. The study was approved by a local ethics committee (Ethic Committee, Medical Faculty, Friedrich-Schiller University, Bachstraße 18, 07743 Jena, Germany; ID: 2019-1401_1-Material). All experimental teeth showed roots that were 12–13 mm in size and of round and straight shape. The crowns were severed at the cervical line using an abrasive cut-off wheel (H Flex, Pluradent; diameter: 19 mm, thickness: 0.15 mm) at 20.000 rpm under constant water cooling. For more convenient handing, the teeth were embedded into casting silicone (Flexitime Easy Putty, HeraeusKulzer GmbH, Hanau, Germany). Subsequently, all root canals were debrided using hand-held files (K-Reamers, K-Files, VDW GmbH, Munich, Germany) using the apical–coronal technique to a standard size of ISO 50. During the mechanical enlargement, the root canals were alternatingly rinsed with 0.9% NaCl solution. For removal of the smear layer, each canal was given a final rinse with 2 mL of 20% EDTA (ethylendiamintetraacid) solution (Calcinase, legeartisPharma GmbH, Dettenhausen, Germany) for 4 min followed by irrigation with 3 mL distilled water. To ensure a leak-proof outer insolation, the root surfaces of the prepared teeth were lined with a 1 mm thick layer of glass ionomer cement (Ketac Bond, 3 M Espe AG, Seefeld, Germany). The prepared teeth were subsequently autoclaved in a moist chamber for 20 min at 121 °C.

### 2.3. Inoculation of Root Canal System with E. faecalis

Ten microliters of the respective *E. faecalis* suspension was transferred to each enlarged (ISO 50) root canal. The batches were then incubated for 48 h at 37 °C under standard anaerobic conditions (Meintrup DVS MK3 Anaerobic Work Station, dw Scientific, Meintrup-Labortechnic, Laenden, Germany). Samples were randomized into 6 test and 2 control groups each with n = 10 specimens. Sample allocation and characterization are summarized in Table 1.

### 2.4. Determination of the Baseline Value

After inoculation for 48 h, root canals were dried twice for 10 s each with ISO 50 paper points (DentsplyDeTrey, Konstanz, Germany). The used paper points (ISO 50) served as an initial colonization control (baseline value) and were traversed to a reaction vessel filled with 1 mL of physiological NaCl. After the establishment of a decadic dilution series to 10^−6^, each aliquot was plated onto Schaedler agar (Schaedler Anaerobic Agar, Oxoid Ltd., Hampshire, UK; 6% sheep’s blood, 0.1% Konakion MM 10 mg, Roche Pharma AG, Basel, Switzerland) and cultivated anaerobically for 3 days. Subsequently, the colony-forming units (CFU/mL) were determined.

### 2.5. Photosensitizer Indocyanine Green (ICG)

For the present investigation, the photosensitizer Indocyanine green (ICG, Figure 1) without a complex to iodine (EmunDo1, A.R.C. Laser GmbH, Nuremberg, Germany) was used. For experimental purposes, final ICG concentrations of 250, 500 and 1000 µg/mL were arranged (Table 1) by dilution of the stock solution (ICG 1 mg; Aqua ad injectabilia 1 mL) with distilled water.

### 2.6. Trolox™

To enhance the photodynamic antibacterial effect of ICG, the vitamin E analogue (+/−)-6-Hydroxy-2,5,7,8-tetramethylchromane- 2-carboxylic acid (Trolox™, Sigma Sigma-Aldrich, St. Louis, MO, USA) was applied. For experimental purposes, final Trolox™ concentrations of 6 mM were arranged by diluting the stock concentration (42 mM) with aqua dest. Two microliters of the Trolox™ solution were added to the respective ICG solutions in test groups 4–6 (Table 1). A structural formula of Trolox™ is shown in Figure 2.

### 2.7. Laser System

A gallium arsenide diode laser (Fox Laser, A.R.C. Laser GmbH, Nuremberg, Germany) at 808 nm (±3 nm) was used for irradiation of the infected dental root canals.

The laser light was transmitted through an optical fiber (Ø 300 mm) that was attached to a tapered endo-applicator with a diffuse radiation end. The laser was operated in continuous wave mode at 0.2 W for 90 s, delivering a light fluence of 100 J/cm^2^.

### 2.8. aPDT Treatment of the Infected Dental Root Canals

After the determination of the baseline values, root canals of all test groups were filled with 10 μL of the respective ICG formulation (Table 1). In test groups 4 to 6 the applied ICG solution was additionally mixed with 2 mL of Trolox™ (6 mM).

After incubation of all test batches in the dark for 10 min, root canals were rinsed with 3 mL PBS and subsequently subjected to laser illumination (0.2 W, 90 s, 100 J/cm^2^).

### 2.9. Controls

Ten infected root canals were rinsed with 3 mL of 3% NaOCl (positive control) and another 10 samples were rinsed with 3 mL physiological NaCl (negative control). All controls are also shown in Table 1.

### 2.10. Collection of Dentin Samples

The method was published previously [7,16]. In brief, following laser or control treatment, each root canal lumen was dried twice for 10 s using ISO 50 paper points (DentsplyDeTrey, Konstanz, Germany), which were then transferred to a reaction vessel filled with 1 mL of normal saline.

The root canal lumen was then mechanically enlarged from ISO 50 to ISO 110 using standardized hand-held files (K-Reamers, VDW GmbH). From one ISO size to the next higher one, there is a 100 μm increase in diameter. Each instrumentation step therefore causes ablation of the dentin wall by approx. 50 µm. The preparation followed a standardized procedure. The canal lumen was shaped using two files of the same ISO size. Each file was turned 10 times by three-quarter rotations until it reached the full working length. After each instrumentation step, the obtained dentin debris was collected into the same Eppendorf reaction vessel. Residual dentin filings were taken up from the root canal lumen by insertion of a paper point of appropriate size for 10 s. The applied paper point was then also transferred to the Eppendorf reaction vessel. After vertexing, serial dilutions up to 10^−6^ were arranged and plated (100 µL) onto blood agar plates. Before proceeding to the next ISO size, the respective root canal was rinsed with 3 mL of sterile physiological NaCl. The described procedure was repeated for each single instrumentation step up to ISO 110. After cultivation of the plated suspensions for 3 days under anaerobic standard conditions, the colony-forming units (CFU/mL) were determined for each single ISO size.

### 2.11. Temperature Profile during NIR Illumination

During intracanal light application, the temperature profile inside the root canal and also at the root surface was recorded. For this purpose, ISO 50 samples were prepared as described above and pre-heated in a water bath to an average temperature of 36 °C. For the temperature recording, no glass ionomer cement lining was applied on the root surface. During illumination (0.2 W, 90 s, 100 Jcm^−1^), the temperature inside the root canal (through the expanded apical dental foramen) and also on the root surface was recorded using thermal sensors (type k, PICO Technology Ltd., Cambridge, UK) connected to a PICO TC08 data logger (also PICO Technology Ltd.).

### 2.12. Statistical Analysis

Significant differences between the test and control groups were determined by applying the Mann–Whitney U test. The *p*-value was set to 0.05. All results were analyzed using SPSS version 24.0.

## 3. Results

Inoculation of the prepared root canals with *E. faecalis* for 24 h led to bacterial colonization of the main root canal walls and also of the dentinal tubules. Analysis of the dentin debris revealed a mean CFU/mL of 6.58 × 10^7^ (baseline value) at the main root canal walls. Furthermore, bacterial colonization of the dentinal tubules up to ISO 110 was observed, which corresponds to a colonization distance of the dentinal tubes >300 µm (Figure 3).

Antibacterial activity was evaluated for ICG in three different concentrations (250, 500, 1000 µg/mL) after illumination with NIR light (808 nm) at a fluence of 100 J/cm^2^.

Compared to the baseline value, aPDT with ICG 250, 500 and 1000 µg/mL resulted in significant bacterial suppression by 2.9, 1.8 and 2.0 log counts (*p* < 0.05) directly at the main root canal walls (Figure 4). In comparison, irrigation with normal saline (negative control) caused a drop in CFU/mL by only 1.1 log counts (*p* < 0.05). From all ICG concentrations tested, aPDT with ICG 250 µg/mL was most efficient.

At size ISO 60, photodynamic treatment with ICG 250 µg/mL resulted in bacterial suppression of 2.7 log counts when compared to the negative control (irrigation with normal saline, *p* < 0.05)). Compared to irrigation with 3% NaOCl (positive control) there was no significant difference in CFU/mL reduction (*p* = 0.255).

Further instrumentation of the main root canal lumen revealed significant bacterial suppression of aPDT with ICG 250 µg/mL up to size ISO 110 when compared to the baseline value (*p* < 0.05). This corresponds to a penetration depth of *E. faecalis* inside the dentinal tubules up to 300 µm. Compared to the positive control (irrigation with 3% NaOCl) no significant difference was observed for aPDT with ICG 250 µg/mL up to ISO 100 (Figure 5).

All other ICG concentrations (500 and 1000 µg/mL) failed to significantly reduce bacterial growth inside the dentinal tubules (Figure 6 and Figure 7). A significant difference to the negative control (irrigation with normal saline) was only found for ICG 500 µg/mL at ISO 60 (*p* = 0.019).

Irrigation with NaOCl (positive control) resulted in complete suppression of *E. faecalis* at the main root canal walls, but also failed to fully eliminate bacteria inside the dentinal tubules. Photodynamic antimicrobial therapy with ICG (250 µg/mL) showed a similar efficiency on bacteria colonizing the dentinal tubules compared to irrigation with NaOCl (Figure 5).

In the present study, the antibacterial effect of a combined photodynamic treatment of ICG, NIR laser light and Trolox™ was also investigated. Irradiation of ICG 250, 500 and 1000 µg/mL in conjunction with Trolox™ (6 mM) caused significant bacterial reduction at the main root canal walls by 2.0, 2.5 and 2.0 log counts (Figure 4). When compared to aPDT without Trolox™, treatment with ICG at a concentration of 250 µg/mL was more efficient (Figure 5). For ICG at a concentration of 500µg/mL, additional application of Trolox™ caused an antibacterial enhancement by 0.7 log counts (Figure 6), while for ICG 1000 µg/mL no difference in bacterial reduction was observed (Figure 7).

The temperature profile inside the main root canal lumen was also recorded during NIR illumination. It was observed that the applied irradiation time and ICG concentration play a major role. An increase in either parameter was directly associated with elevated temperatures.

In detail, applying an output power of 0.2 W (NIR light) in combination with ICG 250, 500 and 1000 µg/mL resulted in temperatures of 28.8 °C, 28.4 °C and 33.6 °C inside the main dental root canal.

Increasing the output power to 0.5 W caused intracanal temperatures of 44.3 °C (250 µg/mL), 43.7 °C (500 µg/mL) and 44.8 °C (1000 µg/mL). All temperature profiles are also presented in Figure 8.

## 4. Discussion

The present in vitro study aimed to investigate the antibacterial photodynamic effect of the photosensitizer (PS) Indocyanine green (ICG) on Enterococcus faecalis in infected dental root canals. The antimicrobial activity of ICG was evaluated for different concentrations (250, 500, 1000 µg/mL) in combination with near-infrared (NIR) laser light (808 nm, 100 Jcm^−2^). As shown by the results, *E. faecalis* was suppressed by a maximum of 2.5 log counts when ICG in a concentration of 250 µg/mL was applied. All other ICG concentrations were of lower antibacterial photodynamic activity.

As proved recently, singlet oxygen production decreases with high Indocyanine green concentrations. This is likely due to the aggregation problems of Indocyanine green at high concentrations. The aggregated molecules cannot absorb the light properly and their photochemical properties change. This causes a decrease in singlet oxygen production [38].

However, other photodynamic systems presented similar results as compared to the present study. In a recent investigation, 5% aminolaevulinic acid in combination with red laser light (635 nm, 50 mW, 7 min) caused bacterial suppression by 3.5 log counts [39]. Using methylene blue (0.1 mg/mL) as PS in conjunction with red laser light (660 nm, 13.2 Jcm^−2^) resulted in a bacterial decrease of 2.21 log [40]. Application of riboflavin (6.25 to 100 μM) and blue laser light (450 nm, 12–30 Jcm^−2^) showed only minor antibacterial effects [41].

In the present investigation, bacterial growth was reduced to a maximum of 2.9 log counts when ICG in a concentration of 250 µg/mL was applied. The efficiency of ICG in suppressing *E. faecalis* has also been observed by other authors. It was reported that illumination of ICG in a concentration of 1000 µg/mL with NIR laser light (810 nm, 31.2 Jcm^−2^) resulted in bacterial growth reduction by 0.74 log [42]. Compared to the results of the present study, aPDT with ICG 1000 µg/mL caused a drop in CFU/mL by 2.0 log counts. In another study, *E. faecalis* in planktonic solution was suppressed by 5.3 log10 when ICG in a concentration of 100 μg/mL was applied [43]. However, in this specific study, a high light fluence (286 Jcm^−2^) was applied which probably caused thermal damage to the microbes. Unfortunately, the authors did not record the temperature during NIR illumination of the culture media.

In this regard, data from our own group has shown that temperature can rise up to 64.5 °C in planktonic suspensions when ICG in a concentration of 500 µg/mL is illuminated by NIR laser light (808 nm) at a fluence rate of 100 Jcm^−2^ [34]. Additional culture analysis revealed that these thermal settings caused a decrease in the viability of *A. actinomycetemcomitans* by 3.7 log counts and resulted in the complete elimination of *F. nucleatum* and *P. gingivalis*. Reduction in ICG (<250 μg/mL) and/or light fluence led to an immediate drop in temperature and thus antibacterial activity [34]. In a recent study, recordings with a thermal camera showed that near-infrared light irradiation of ICG (6.3 µg/cm^2^) increased the temperature of the surrounding medium by >44 °C [44].

However, in the present study the temperature inside the main root canal did not exceed 48.8 °C. This value was recorded during aPDT with 1000 µg/mL ICG and NIR laser light at an output power of 0.5 W. Reduction in the power to 0.2 W resulted in intracanal temperatures that ranged between 28.8 °C (ICG 250 µg/mL) and 33.6 °C (ICG 1000 µg/mL). Since a power setting of 0.2 W was maintained throughout the entire study, significant thermal damage to *E. faecalis* can be excluded.

As shown by our group in planktonic culture already, treatment with ICG alone only showed a significant dark toxic effect on *P. gingivalis* when high concentrations (>250 µg/mL) were applied. Likewise, irradiation with NIR laser light in the absence of ICG did not cause any significant antibacterial effect or heating of the bacterial suspension. Trolox™ administrated alone or in combination with NIR laser light showed no suppressive effect. In the absence of ICG, irradiation of a Trolox™-containing solution with NIR laser light (100 J/cm^2^) also caused no heating of the bacterial suspension [34].

However, ICG-based photodynamic therapy shows cytotoxic effects also on mammalian cells. It was documented that NIR light at an energy density of 84 J/cm^2^ was quite safe for keratinocytes with ICG concentrations ranging from 4 to 125 μg/mL. When 252 J/cm^2^ energy density was applied, most of the keratinocytes were damaged with any photosensitizer concentration. Fibroblasts tolerated these energy densities only to ICG concentrations of 10 μg/mL. Increasing the photosensitizer concentration resulted in high phototoxic effects [45]. Another study indicated that ICG after angiography induces potential phototoxicity on human retinal pigment epithelial cells via oxidative damage under continuous ambient illumination. Cytotoxicity was reduced by blocking green to red light wavelengths [46]. Further investigations found a cytoplasmic distribution of ICG, probably bound to glutathione S-transferase. Following irradiation with a CW diode laser (805 nm, 80 mW/cm^2^) at doses of 24 or 48 J/cm^2^, ICG concentrations >25 µM produced a significant phototoxic effect [47]. The combined effect of ICG and Trolox™ on mammalian cells still needs to be evaluated.

In the present study, a maximum of 2.9 log in reduction was observed for aPDT with 250 µg/mL ICG. The photodynamic effect of ICG in endodontic disinfection has been discussed also by other authors. In a recent study, treatment of infected root canals with a combination of ICG (1000 µg/mL), silver nanoparticles and NIR laser light (808 nm, 250 mW, 60 s) caused suppression of *E. faecalis*, but only by 0.5 log CFU/mL [48]. Another current investigation also confirmed a rather low antibacterial effect of ICG on *E. faecalis* in infected dental root canals [49]. Stronger antibacterial activity was received when ICG-loaded nanospheres coated with chitosan were applied. Irradiation of the root surface at 2.1 W for 5 min resulted in a reduction of *E. faecalis* by 1.89 log, which was similar to the findings of the present study (ICG 500 and 1000 µg/mL) [50]. Other authors observed sufficient antibacterial effects when ICG was conjugated to nano-curcumin and metformin or by applying more sufficient laser settings [29,51].

In the present examination, a different approach was addressed for enhancing the photodynamic antibacterial activity of ICG. As shown by previous results of our group, an additional application of the vitamin E analogue Trolox™ can significantly enhance the effect of ICG-based aPDT [34].

But, in the present investigation, additional application of Trolox™ did only cause a minor enhancing effect. Only in the case of the 500 µg/mL ICG concentration, bacterial reduction increased from 1.8 to 2.5 log when Trolox™ was additionally added to the root canal system. The defined null hypothesis can therefore only partly be rejected.

The enhancing potential of vitamin E (α-tocopherol) in photodynamic therapy has been discussed also by other authors. As shown, the application of α-tocopherol resulted in significantly increased photoinactivation of HT 29 adenocarcinoma cells [36]. Moreover, a delay in tumor doubling time from 13 to 19 days was observed when Trolox™ was injected in tumor-bearing mice at concentrations of 250 mg/kg body weight 90 minutes prior to the photodynamic intervention [52]. In this regard, results from our group showed that in the presence of Trolox™, ICG can be reduced to one-fifth of its original concentration, still causing complete photodynamic suppression of Fusobacterium nucleatum and Porphyromonas gingivalis [34].

Besides vitamin E, it is known that other antioxidants, such as ascorbate, β-carotene and 3(2)-tert-butyl-4-hydroxyanisole at adequate concentrations also present enhancing photodynamic characteristics [53,54,55,56].

However, antioxidants normally function as scavengers and neutralizers of reactive oxygen radicals (ROS). But, under certain circumstances, those antioxidants can also present pro-oxidant activity.

Those characteristics mainly arise from an excessive formation of antioxidant radicals due to either spontaneous autooxidation or radical scavenging during the photodynamic reaction [36]. It was also observed that antioxidants serve as substrate for photosensitizers in the excited triplet state. This causes the formation of radical photosensitizer anions and antioxidant radicals which might further react with residual oxygen leading to the formation of superoxide radical anions and other reactive molecules [37].

In the present study, the application of Trolox™ showed only minor enhancing effects. A reason might be that the antioxidative activity of Trolox™ predominates in the present setting, scavenging radicals that originate from the photodynamic reaction. This is also confirmed by results showing that Trolox™ is a powerful agent for preventing oxidative cell damage mediated by resin-containing dental restorative materials [57,58].

Besides lacking information on the combined photodynamic action of ICG and Trolox™ on mammalian cells, further limitations of the present study can be seen in the NIR system. In the present study, the applied laser parameters and irradiation setup might have failed to deliver sufficient quantities of light to the root canal system. Furthermore, the limited availability of molecular oxygen in the root canals can also be seen as a further reason for the low efficiency. There is probably also an interaction between ICG, Trolox™ and the root canal dentin which might reduce the photodynamic activity. A further limitation of the study can be seen in the point that the effect of ICG, Trolox™ and NIR light alone was not exclusively addressed. But, as already discussed above, no extensive cytotoxic activity is expected.

In this regard, future studies should especially focus on the enhancing characteristics of Trolox™, also in combination with other photosensitizers. The not consistently detectable effect of Trolox™ that was evaluated in the present investigation might be referred to the specific root canal environment and can be further influenced by contact with dentin. The results of the present study are less efficient since strong enhancing characteristics of Trolox™ were already detected in the planktonic culture [34]. Compared to irrigation with NaOCl, aPDT with ICG showed similar results on bacteria that colonized the dentinal tubules. It can be assumed that aPDT generally presents fewer side effects than treatment with NaOCl [18,20,59]. In total, aPDT with ICG can be seen as a potential antimicrobial measure in the adjunct treatment of recurrent endodontic infections.

## 5. Conclusions

In the present investigation, it was shown that photodynamic treatment of infected dental root canals with ICG was efficient in suppressing *E. faecalis*. An additional application of Trolox™ caused minor enhancing effects. It was further investigated that application of NIR laser-light is safe and does not cause extensive thermal heating, when applied in the proposed settings. It can be concluded that the method effectively suppresses bacteria inside dentinal tubules with comparable results to NaOCl, but possibly with fewer side effects.

## Figures and Tables

**Figure 1 pharmaceutics-15-02572-f001:**
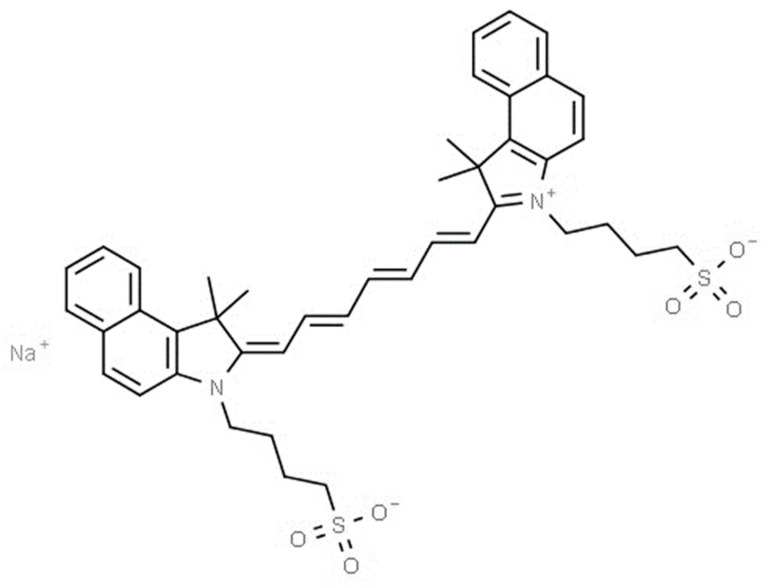
Structural formula of Indocyanine green (ICG). Image was adopted from www.chemspider.com (accessed on 12 October 2023).

**Figure 2 pharmaceutics-15-02572-f002:**
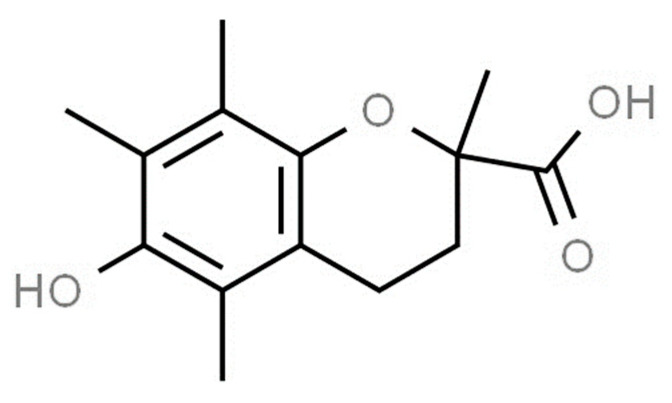
Structural formula of Trolox™. Image was adopted from www.chemspider.com (accessed on 12 October 2023).

**Figure 3 pharmaceutics-15-02572-f003:**
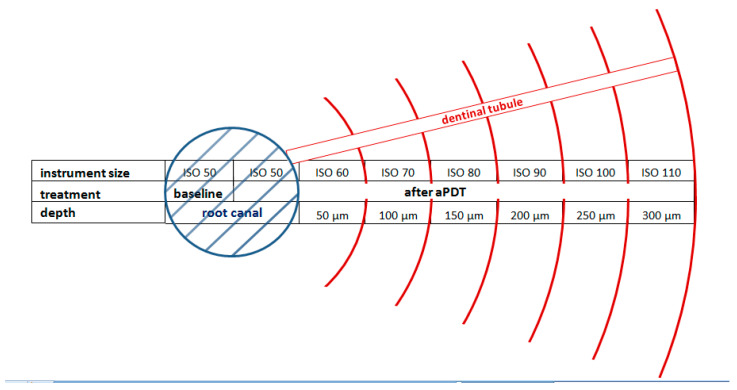
Schematic drawing showing the relation between mechanical instrumentation of the root canal lumen (ISO size) and bacterial penetration depth. Root canal size at baseline is shown in blue. Subsequent mechanical preparation and distance of bacterial invasion into the dentinal tubules is shown in red. Image was adopted from [16] (Permission has been obtained from Dr. Ossmann, Copyright 2014, Springer-Verlag Berlin Heidelberg, Germany).

**Figure 4 pharmaceutics-15-02572-f004:**
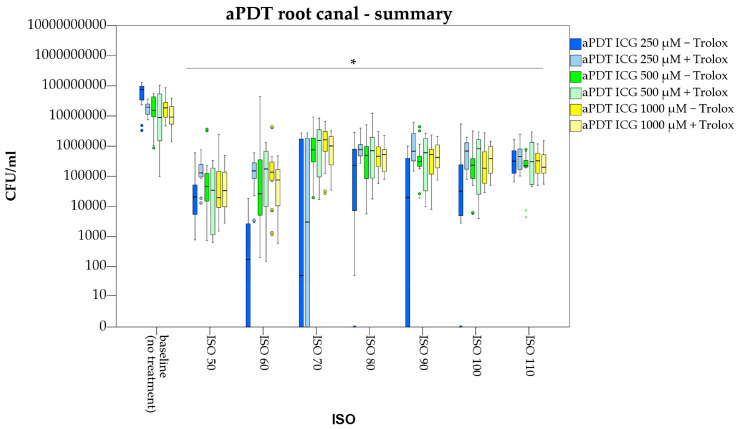
Photodynamic antimicrobial treatment of infected dental root canals with Enterococcus faecalis. For aPDT the photosensitizer Indocyanine green (ICG) with or without Trolox™ was used Root canals were manually enlarged from ISO 50 to ISO 110. Star indicates significance in CFU/mL (ISO size) to the respective baseline value (*p* < 0.05). Colored circles and stars indicate outliers.

**Figure 5 pharmaceutics-15-02572-f005:**
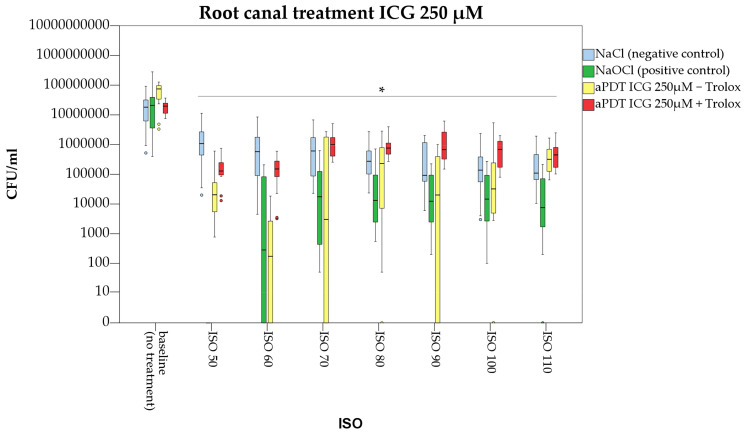
Photodynamic treatment of infected dental root canals with Indocyanine green (ICG) in a concentration of 250 µg/mL. Trolox™ was additionally applied. Root canals were manually enlarged from ISO 50 to ISO 110. Star indicates significance in CFU/mL (ISO size) to the respective baseline value (*p* < 0.05). Colored circles indicate outliers.

**Figure 6 pharmaceutics-15-02572-f006:**
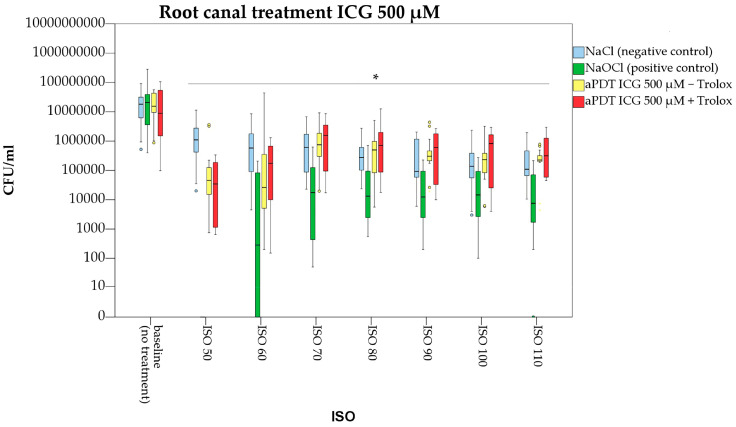
Photodynamic treatment of infected dental root canals with Indocyanine green (ICG) in a concentration of 500 µg/mL. Trolox™ was additionally applied. Root canals were manually enlarged from ISO 50 to ISO 110. Star indicates significance in CFU/mL (ISO size) to the respective baseline value (*p* < 0.05). Colored circles indicate outliers.

**Figure 7 pharmaceutics-15-02572-f007:**
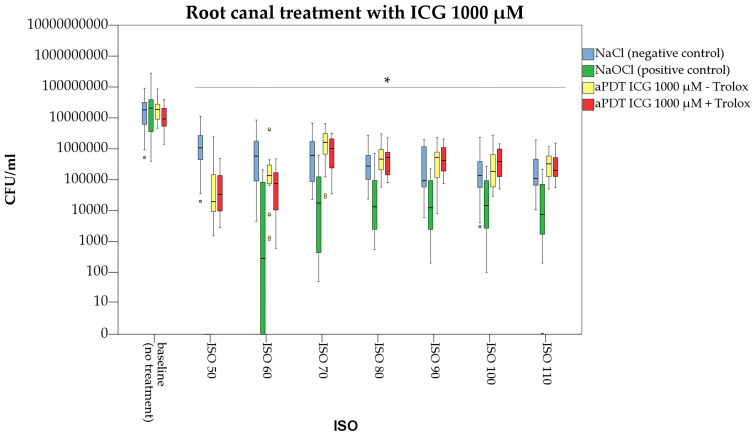
Photodynamic treatment of infected dental root canals with Indocyanine green (ICG) in a concentration of 1000 µg/mL. Trolox™ was additionally applied. Root canals were manually enlarged from ISO 50 to ISO 110. Star indicates significance in CFU/mL (ISO size) to the respective baseline value (*p* < 0.05). Colored circles indicate outliers.

**Figure 8 pharmaceutics-15-02572-f008:**
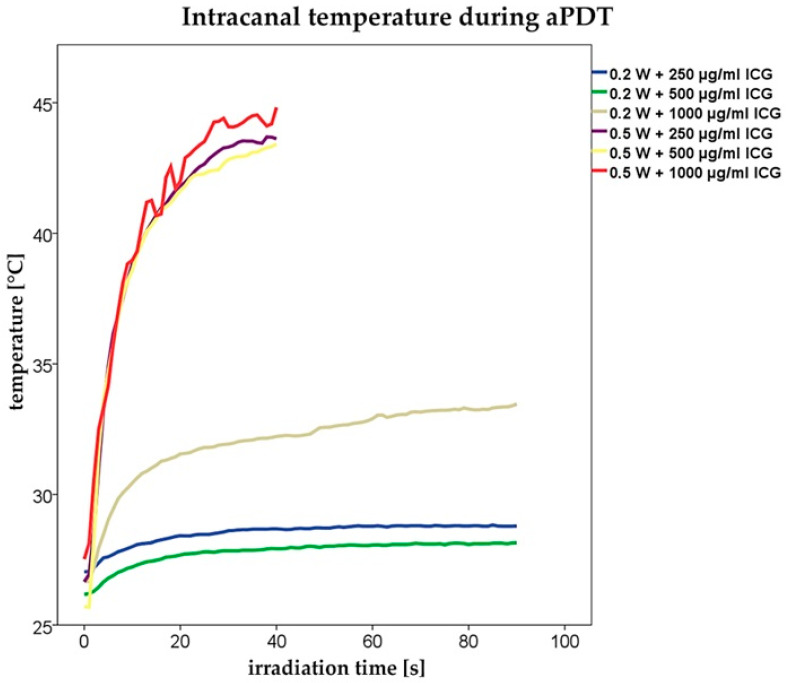
Intracanal temperature profiles during NIR illumination with different ICG concentrations and power settings.

**Table 1 pharmaceutics-15-02572-t001:** Sample allocation.

	Number of Samples	ICG-Concentration [µg/mL]	Trolox^TM^ [6 mM]	NIR Laser (808 nm, 100 Jcm^−2^)	NaOCl [mL]	NaCl [mL]
Test group 1	10	250	-	+	0	0
Test group 2	10	500	-	+	0	0
Test group 3	10	1000	-	+	0	0
Test group 4	10	250	+	+	0	0
Test group 5	10	500	+	+	0	0
Test group 6	10	1000	+	+	0	0
Positive control	10	0	-	-	3	0
Negative control	10	0	-	-	0	3

## Data Availability

Data can be obtained from the corresponding author upon request.

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
