# Peer review of "Photodynamic Suppression of Enterococcus Faecalis in Infected Root Canals with Indocyanine Green, TroloxTM and Near-Infrared Light"

_pharmaceutics, 2023, doi:10.3390/pharmaceutics15112572_

Round 1

Reviewer 1 Report

Comments and Suggestions for Authors

Markus Heyder et al. detected the antimicrobial  photodynamic  effect  of ICG and  Trolox™  on  Enterococcus  faecalis.

As a result, it was found that  ICG significantly suppressed E. faecalis.

aPDT may have certain potential in treatment of endodontic diseases.

Major points:

1. The cytotoxicity of the present aPDT combination to mammalian oral cells is not mentioned.

2. If possible, the other periodontal clinical indices, like probing dpth, plaque index, can also be detected.

3. clinical experiment or animal experiment can be performe to assess the effect of the present aPDT in vivo.

4. Does the  present aPDT has effects on other endodontic bacterial species?

Comments on the Quality of English Language

The manuscript is well written.

Author Response

Dear reviewer,

thank you for revising our manuscript. We tried to answer each comment to the best of our knowledge. All changes among the manuscript are highlighted in yellow. Please find below the answers to your comments.

Again, many thanks for your valuable time!

Sincerely yours,

Dr. Stefan Kranz

Markus Heyder et al. detected the antimicrobial photodynamic effect of ICG and Trolox™ on Enterococcus faecalis. As a result, it was found that ICG significantly suppressed E. faecalis. aPDT may have certain potential in treatment of endodontic diseases.

Comment: The cytotoxicity of the present aPDT combination to mammalian oral cells is not mentioned.

Answer: Thank you for this remark. We included information about the cytotoxicity of ICG on mammalian cells. The combined effect of ICG and Trolox™ still needs to be evaluated. Please view lines 326 to 339.

Comment: If possible, the other periodontal clinical indices, like probing depth, plaque index, can also be detected.

Answer: In the present study we just focused on endodontic disinfection of root canals infected by E. faecalis in-vitro. In periodontal studies, pocket depth, attachment level and BoP are relevant clinical indices.

Comment: Clinical experiment or animal experiment can be performed to assess the effect of the present aPDT in vivo.

Answer: Thank you for this remark. Our group is currently planning on a large animal study observing the antimicrobial effects of different photodynamic systems. ICG will be incorporated into the experimental set up. Results of these studies will be published in near future.

Comment: Does the present aPDT has effects on other endodontic bacterial species?

Answer: It was shown by various studies that aPDT with ICG is efficient in suppressing different grampositive and gramnegative bacteria. Probably, ICG is also efficient on other endodontic species. As shown in doi: 10.1016/j.pdpdt.2014.12.006. photodynamic treatment with ICG is even capable in suppressing Candida albicants. For more detailed information on the antibacterial activity of ICG please refer to doi: 10.3390/molecules28166085. 

Reviewer 2 Report

Comments and Suggestions for Authors

Interesting and well performed study on the effects of different combined therapies on the growth of E fecalis in the endodontic field. Just some criticisms listed below:

Some criticisms reported below:

-An initial sentence in the abstract section relating to the problem that led to the study must be inserted

-check that all keywords are PUBMED MESH terms

-in the introduction section some considerations should be added on the microscopic effects of a post-endodontic periapical lesion. In this regard, I ask you to include the following scientific work in the reference section which could be of help to the authors:

Chieruzzi M, Pagano S, De Carolis C, Eramo S, Kenny JM. Scanning Electron Microscopy Evaluation of Dental Root Resorption Associated With Granuloma. Microsc Microanal. 2015 Oct;21(5):1264-70. doi: 10.1017/S1431927615014713. Epub 2015 Aug 3. PMID: 26235380.

- insert the null hypotheses of the study at the end of the introduction section which will then be refuted in light of the results obtained

- some considerations and bibliographical references are also missing on the effect of laser therapy in combination with traditional therapy in endodontic therapy.

-Add in the introduction section some considerations on the potential mechanisms underlying the effects of TROLOX

-why were 10 samples per group selected? Indicate it explicitly

-A section on the limitations of the study is missing

-Check that all references are written as editorial indications

Comments on the Quality of English Language

moderate check must be added

Author Response

Dear reviewer,

thank you for revising our manuscript. We tried to answer each comment to the best of our knowledge. All changes among the manuscript are highlighted in green. Please find below the answers to your comments.

Again, many thanks for your valuable time!

Sincerely yours,

Dr. Stefan Kranz

Interesting and well performed study on the effects of different combined therapies on the growth of E fecalis in the endodontic field. Just some criticisms listed below:

Comment: An initial sentence in the abstract section relating to the problem that led to the study must be inserted

Answer: Thank you for this remark. An additional sentence was introduced which resulted in some further chances among the first paragraph in the abstract.

Comment: Check that all keywords are PUBMED MESH terms

Answer: All keywords were checked and modified if necessary.

Comment: In the introduction section some considerations should be added on the microscopic effects of a post-endodontic periapical lesion. In this regard, I ask you to include the following scientific work in the reference section which could be of help to the authors:

Chieruzzi M, Pagano S, De Carolis C, Eramo S, Kenny JM. Scanning Electron Microscopy Evaluation of Dental Root Resorption Associated With Granuloma. Microsc Microanal. 2015 Oct;21(5):1264-70. doi: 10.1017/S1431927615014713. Epub 2015 Aug 3. PMID: 26235380.

Answer: The recommended information was introduced to the introduction. Chieruzzi et al. was added to the references.

Comment: Insert the null hypotheses of the study at the end of the introduction section which will then be refuted in light of the results obtained

Answer: Thank you for this remark. We introduced a null-hypothesis to the introduction. Further information regarding the null-hypothesis were also discussed. Please view lines 87 to 88 as well as 359.

Comment: Some considerations and bibliographical references are also missing on the effect of laser therapy in combination with traditional therapy in endodontic therapy.

Answer: We introduced further information about the issues. Please refer to lines 52 to 55.

Comment: Add in the introduction section some considerations on the potential mechanisms underlying the effects of TROLOX

Answer: Thank you for this comment. We introduced further information upon the proposed enhancing effect of Trolox™. Please view lines 76 to 82.

Comment: Why were 10 samples per group selected? Indicate it explicitly

Answer: We chose the number of samples because of statistical reasons. For more detailed information please also refer to DOI 10.1007/s00784-014-1271-9 and DOI 10.3390/ma14092427.

Comment: A section on the limitations of the study is missing

Answer: Thank you for this comment. We introduced further information regarding the limitations of the study. Please view lines 386 to 395.

Comment: Check that all references are written as editorial indications

Answer: Reference style was checked. MDPI Endnote style was used.

Reviewer 3 Report

Comments and Suggestions for Authors

The main point of the manuscript “Photodynamic suppression of Enterococcus faecalis in infected root canals with indocyanine green, TroloxTM and near-infrared light” is the known antibacterial effect of a combination of a photosensitizer and an antioxidant. In this work, the well-studied photosensitizer indocyanine green and a vitamin E analog Trolox were employed to study the possibility of therapy of root canals infected with E. faecalis. Experimental work is done at a good level, all studies are executed well in terms of research methodology and data curation.

However, the authors provide no explanation for the apparent lack of dose-dependent effect of the photosensitizer both with and without Trolox. For instance, with Trolox, maximum activity is achieved at 500 mkg/mL of ICG, lower and higher concentrations being worse, and without Trolox, the lowest concentration of ICG works best – why could that be?

Additional controls are required: a control consisting of laser treatment without photosensitizer, as well as a control with laser and Trolox but no ICG.

The article also lacks a more detailed discussion of the results obtained. What are the advantages and disadvantages of the approach, taking into account that the effect is inferior to or on par with that of existing drugs (e.g., NaOCl)?

Minor issues:

There are some comments on the text of the article: a typo in the title of the article “indocynine” instead of “indocyanine”, and also the species name of E. faecalis in italics throughout: lines - 14, 34, 36, 40, 41, 45, 58, 75, 77, 81, 105, 187, 212, 262, 265, 274, 278, 297, 302, 303, 306, 357, the same in italics 285-287 A. actinomycetemcomitans, F. nucleatum, P. gingivalis, line 225 and 259 should be shortened to E. faecalis.

Structures of compounds should be probably added for better comprehension.

Author Response

Dear reviewer,

thank you for revising our manuscript. We tried to answer each comment to the best of our knowledge. All changes among the manuscript are highlighted in light blue. Please find below the answers to your comments.

Again, many thanks for your valuable time!

Sincerely yours,

Dr. Stefan Kranz

The main point of the manuscript “Photodynamic suppression of Enterococcus faecalis in infected root canals with indocyanine green, TroloxTM and near-infrared light” is the known antibacterial effect of a combination of a photosensitizer and an antioxidant. In this work, the well-studied photosensitizer indocyanine green and a vitamin E analog Trolox were employed to study the possibility of therapy of root canals infected with E. faecalis. Experimental work is done at a good level, all studies are executed well in terms of research methodology and data curation.

Comment: However, the authors provide no explanation for the apparent lack of dose-dependent effect of the photosensitizer both with and without Trolox. For instance, with Trolox, maximum activity is achieved at 500 mkg/mL of ICG, lower and higher concentrations being worse, and without Trolox, the lowest concentration of ICG works best – why could that be?

Answer: Thank you for this important comment. As proved recently, singlet oxygen production decreases with high indocyanine green concentrations. This is likely due to the aggregation problems of indocyanine green at high concentrations. The aggregated molecules cannot absorb the light properly and their photochemical properties change. This causes a decrease in singlet oxygen production. We addressed this issue in the discussion. Please view lines 282 to 286.

Comment: Additional controls are required: a control consisting of laser treatment without photosensitizer, as well as a control with laser and Trolox but no ICG.

Answer: Thank you for this important comment. As shown by our group in planktonic culture already, treatment with ICG alone only showed a significant dark toxic effect on P. gingivalis when high concentrations (> 250 µg/ml) were applied. Likewise, irradiation with NIR laser light in the absence of ICG did not cause any significant antibacterial effect or heating of the bacterial suspension. Trolox™ administrated alone or in combination with NIR laser light showed no suppressive effect. In the absence of ICG, irradiation of a Trolox™ containing solution with NIR laser light (100 J/cm2) also caused no heating of the bacterial suspension. We introduced this information to the discussion section. Please view lines 319 to 325. Cytotoxicity of ICG, Trolox™ and NIR light was also investigated in planktonic culture on E. faecalis. Results can be obtained on request from the corresponding author. In the present study there was no additional testing of these parameters in the applied root canal model. We therefore addressed these issues in the discussion section. Please view lines 392 to 395.

Comment: The article also lacks a more detailed discussion of the results obtained. What are the advantages and disadvantages of the approach, taking into account that the effect is inferior to or on par with that of existing drugs (e.g., NaOCl)?

Answer: The results show that aPDT with ICG +/- Trolox™ efficiently eliminates bacteria in the dental root canal and especially of microbes that colonize the dentinal tubules. The effect was comparable to irrigation with 3% NaOCl. Compared to irrigation with NaOCl of higher concentration (>2.5%), application of aPDT shows fewer side effects, too. As also addressed in the publication, aPDT should therefore be seen as an adjunct treatment measure. These are the main advantages and have already been discussed in the present study. Please view lines 401 to 413. The main disadvantages are probably due to the methodology of aPDT in the root canal system. Sufficient application of light and photosensitizer is crucial and sometimes limited by the root canal anatomy. These issues have been addressed also in the present study. Please refer to lines 56 to 70.

Comment: There are some comments on the text of the article: a typo in the title of the article “indocynine” instead of “indocyanine”, and also the species name of E. faecalis in italics throughout: lines - 14, 34, 36, 40, 41, 45, 58, 75, 77, 81, 105, 187, 212, 262, 265, 274, 278, 297, 302, 303, 306, 357, the same in italics 285-287 A. actinomycetemcomitans, F. nucleatum, P. gingivalis, line 225 and 259 should be shortened to E. faecalis.

Answer: Thank you for this remark. All species names are in italic now.

Comment: Structures of compounds should be probably added for better comprehension.

Answer: Structural formulas of ICG and Trolox™ are included. Please view Figures 1 and 2.

Round 2

Reviewer 3 Report

Comments and Suggestions for Authors

Authors addressed all the comments from referees. The manuscript is now suitable for publication.